# The Financialization of Crude Oil Markets and Its Impact on Market Efficiency: Evidence from the Predictive Ability and Performance of Technical Trading Strategies

**Cristiana Tudor [1],\* and Andrei Anghel [2]**

1   International Business and Economics Department, The Bucharest University of Economics, 010374 Bucharest, Romania
2   CEO, Finpathic—Roboadvisor, 040672 Bucharest, Romania; anghel.finpathic@gmail.com
\*   Correspondence: cristiana.tudor@net.ase.ro

**Abstract:** Oil price forecasts are of crucial importance for many policy institutions, including the European Central Bank and the Federal Reserve Board, but projecting oil market evolutions remains a complicated task, further exacerbated by the financialization process that characterizes the crude oil markets. The efficiency (in Fama's sense) of crude oil markets is revisited in this research through the investigation of the predictive ability of technical trading rules (TTRs). The predictive ability and trading performance of a plethora of TTRs are explored on the crude oil markets, as well as on the energy sector ETF XLE, while taking a special focus on the turbulent COVID-19 pandemic period. We are interested in whether technical trading strategies, by signaling the right timing of market entry and exits, can predict oil market movements. Research findings help to confidently conclude on the weak-form efficiency of the WTI crude oil and the XLE fund markets throughout the 1999–2021 period relative to the universe of TTRs. Moreover, results attest that TTRs do not add value to the Brent market beyond what may be expected by chance over the pre-pandemic 1999–2019 period, confirming the efficiency of the market before 2020. Nonetheless, research findings also suggest some temporal inefficiency of the Brent market during the 1 and $\frac{1}{4}$ years of pandemic period, with important consequences for energy markets' practitioners and issuers of policy. Research findings further imply that there is evidence of a more intense financialization of the WTI crude oil market, which requires tighter measures from regulators during distressed markets. The Brent oil market is affected mainly by variations in oil demand and supply at the world level and to a lesser degree by financialization and the activity of market practitioners. As such, we conclude that different policies are needed for the two oil markets and also that policy issuers should employ distinct techniques for oil price forecasting.

**Keywords:** crude oil; energy markets; technical trading rules; predictability; data snooping; market efficiency; COVID-19 pandemic

## 1. Introduction

Over the last decades, the oil market registered significant growth, becoming the world's biggest commodity market and transforming from a purely physical to a highly sophisticated and complex financial market [1]. Its rhythm of growth remains high: the global oil and gas market is expected to grow from \$4677.45 billion in 2020 to \$5870.13 billion in 2021 at a compound annual growth rate (CAGR) of 25.5%, and the market is expected to reach \$7425.02 billion in 2025 at a CAGR of 6% [2]. In addition, crude oil also tops the commodities markets in terms of liquidity, being the most actively traded commodity around the globe, while the price of oil reflects the overall health of the energy sector worldwide.

Oil price forecasts are of crucial importance for macroeconomic projections, which is especially explained by the impact that oil prices have on inflation and output and,

consequently, on the issuance of monetary policy. However, recent movement of crude oil markets has highlighted the difficulty in forecasting oil prices and attested that oil market dynamics tend to vary substantially over time. Moreover, crude oil markets are characterized by increased volatility, which might be explained both by variations in the price elasticity of oil demand and supply, and also by the process of "financialization" of the oil market with the increasing use of oil as a financial asset [3–6]. Consequently, oil derivatives markets have expanded over the last decades, with the presence of purely financial practitioners (institutional investors such as hedge funds, pension funds, insurance companies, and also individual traders) with no interest in the physical commodity becoming more prominent. Concurrently, a variety of instruments that permit speculation in oil have become available for trading, from passive investment vehicles such as energy indexes and ETFs to derivative instruments such as futures, options, or CDFs. All these developments in oil markets have a direct impact on the oil market movements, its efficiency and subsequent predictability.

Financial institutions and regulators around the globe (i.e., The Federal Reserve Board, the World Bank, the International Energy Agency, the European Central Bank etc.) regularly issue oil price forecasts, which is further a paramount factor for policy formulation within the European Central Bank (ECB), the IMF and the Federal Reserve Board [7]. However, predicting oil price movements remains a challenging endeavor [3], which is further complicated by its increasing financialization and the intense speculative activity within the market that improved its efficiency (in Fama's EMH sense) and hence contributed to its unpredictability. Moreover, none of the techniques previously employed for oil price forecasting has proved particularly successful and thus presently there is no "optimal" or commonly accepted forecasting technique for oil price [8].

As such, the analysis of the efficiency of the crude oil markets is a timely research topic, with important implications for policy issuers and for financial markets practitioners. Nonetheless, and somewhat surprisingly given the practitioners' interest in this commodity as reflected in its market liquidity, the academic literature on the profitability of technical trading rules applied to crude oil markets remains rather scarce. Our study contributes to extending this literature. This paper thus revisits the Fama efficiency [9] of the crude oil markets though exploring the predictive ability and trading performance of a plethora of technical trading rules (TTRs) applied to relevant energy series (i.e., WTI crude, Brent crude and XTE). Moreover, our focus on energy/oil markets is even more motivated by the fact the COVID-19 pandemic has severely impacted the oil markets, due to travel restrictions, disrupted supply chains and imposed government lockdowns. Previous studies have found that the efficiency of crude oil markets is lost during crisis periods, investigating the 2008 global financial crisis [10,11]. As the impact of the ongoing pandemic crisis on the oil market efficiency has not been yet assessed, this constitutes a secondary research goal of the current study and a further contribution to the extant literature.

The Efficient Market Hypothesis (EMH) and the related concept of market efficiency remain paramount in modern finance, with a plethora of empirical studies dedicated to confirm it on different markets, assets and time periods, with divergent results. EMH has its roots in the works of Eugene Fama [12,13] and Paul Samuelson [14]. Furthermore, Fama's seminal work defines an efficient market as "a market with a large number of rational, profit "maximisers" actively competing, each attempting to predict future market values of individual securities, and where current important information is almost freely available to all participants" [15] and it also distinguishes between three forms (or 'strengths') of market efficiency—weak, semi-strong and strong. In its weak-form, EMH states that current prices reflect all existing historical information, and thus prices will exhibit random walk.

Alternatively, technical analysis (or Chartism) specifically involves making investment decisions based on past price movements. As Alexander [16] has said it, "the technician studies price movements of the immediate past for telltale indications of the movements of the immediate future." However, in relation to EMH, this would imply that technical trading rules (TTRs) based on historical price data would offer no predictive power, and

hence technical analysis would be inexpedient. Nonetheless, as Menkhoff [17] shows, technical analysis remains very popular among practitioners, with the vast majority of 692 surveyed fund managers from five countries acknowledging relying on technical analysis for market timing and decision-making, and to favor it relative to fundamental analysis. This is an indication that Chartism must hold some value to traders that is unaccounted by the EMH.

Consequently, in this study, we choose to employ instruments pertaining to technical analysis (i.e., TTRs) to investigate the overall efficiency of the oil markets, to assess the potential differing financialization process of the two most important crude oil markets (WTI and Brent) and to analyze the impact of the COVID-19 induced crisis on oil markets' efficiency.

The remainder of the paper is organized as follows. The next section gives a review of the literature concerned with technical trading rules applied to commodity markets and most specifically their predictive ability and performance on oil/energy markets. Section 3 discusses the data and method. Empirical results and discussions are contained in Section 4, followed by some concluding comments in the final section.

## 2. Literature Review

Although the predictive ability and profitability of technical trading rules applied to various international stock markets during different time periods have been thoroughly examined, the literature on technical trading rules applied to commodities markets in general and energy/oil markets in particular remains rather scarce.

One of the first studies in this narrow literature is that of Marshall et al. [18], which test over 7000 rules on 15 commodity futures markets, including WTI crude oil, heating oil and soybean oil for a period spanning 1 January 1984–31 December 2005. They analyze the entire series and two equal sub-periods and cannot report that technical rules achieve superior performance after accounting for data snooping, except the oats market. The oil markets are thus found to be efficient over the 1984–2005 period.

Further, Szakmary et al. [19] examine the profitability (net of transactions costs) arising from the implementation of 12 trading rules (six DMAC and six channel specifications) on a monthly dataset for 28 commodities, having a different start date for each series and with all series ending on 31 December 2007. The dataset includes the same three oil markets again, i.e., WTI crude oil, heating oil and soybean oil, and results confirm that technical rules do perform well, although mean returns are lower and less significant toward the end of the analyzed period (i.e., during the 1996–2007 sub-period), especially when testing is restricted to high-volume markets, a category to which WTI crude oil belongs. Nonetheless, the authors refrain to claim that their study confirms the weak-form inefficiency of commodity futures markets included in the analysis.

Narayan et al. [20] use daily data on four commodities, including again WTI crude oil, spanning the period 16 May 1983–22 November 2011, to which they apply a narrow universe of six standard moving average (SMA) trading rules and report that investors can earn abnormal return (net of commissions) from technical trading rules in three of the four markets, including in the WTI crude oil market, where trading rules achieve the highest return. However, their results do not seem to account for data snooping, which is a bias proven to have a significant impact on results and thus are not sufficiently strong to prove the inefficiency of the WTI crude oil market. Subsequently, Narayan et al. [21] also conclude that commodity futures markets can indeed offer investors statistically significant profits.

Further, Wang et al. [22] employ daily prices of WTI crude oil futures contracts over 1983–2014 and develop dynamic MA trading strategies through genetic algorithms, whose trading performance is further compared to the buy-and-hold strategy and to some static MA rules. The study confirms the superiority of dynamic moving averages on the WTI crude oil futures market during downward trending markets. However, it also lacks a check of results robustness.

More recently, Psaradellis et al. [23] offer probably the most updated study on technical trading rules applied to the crude oil market. The study thus investigates the success of the 7846 trading rules proposed by Sullivan et al. [24] applied on the daily prices of WTI crude oil futures and on the United States Oil (USO) fund, from 2006 to 2019. Results confirm that there is no persistent nature in rules' performance for the two oil markets after adjusting for data snooping, thus supporting WTI market efficiency for the 2006–2019 period, although some interim market inefficiencies might be encountered.

Overall, previous studies thus generally agree on the efficiency of the WTI crude oil market for different periods, all spanning before the ongoing COVID-19 pandemic, after adjustment for data snooping-bias is made. To the best of our knowledge, the efficiency of the Brent crude oil market in relation to the performance of technical trading strategies has not been tested, nor has the efficiency of the XLE fund market. This study intends to fill this void, providing relevant results for policy makers, academics and investment practitioners.

Thus, we add to the literature first by extending the energy markets under scrutiny by including the most traded crude oil contract at world level, i.e., Brent crude oil along with a relevant energy-traded ETF, namely XLE and, secondly and most importantly, by an updated investigation on the performance of a large universe of TTRs during an historically turbulent period for crude oil markets and energy portfolios (i.e., the COVID-19 pandemic).

Additionally, a non-trivial issue about TTRs and their performance that needs further discussion is testing the statistical significance of results.

In this respect, bootstrapping firstly emerged as a convenient way of testing TTRs on data generated using some algorithm. Brock, Lakonishok and LeBaron [25] proposed the bootstrapping methodology for testing the predictability of some of the simplest trading rules and found that technical rules—in particular SMA—were able to achieve excess returns that could not be explained by a random walk model, an AR (1) process, nor a GARCH (M or Exponential) model. Another method, the stationary bootstrap that resamples from blocks of data with random lengths, was developed by Politis and Romano [26]. However, the bootstrapping methodology developed by Brock et al. [25] is the one that has been extensively applied in the literature concerned with the profitability and predictability of TTRs on speculative markets.

Nonetheless, this method is vulnerable to the so-called data-snooping bias. Data snooping reflects the process of testing and retesting filters, rules and combinations on a high number of randomly generated series until some (apparently) significant specifications emerge. In other words, the data snooping bias reflects the danger that the best forecasting model encountered in a specification search is just the result of chance instead of superior forecasting abilities and thus has no predictive superiority over a given benchmark model. Among others, Fang et al. [27] demonstrate that the predictive ability of the technical trading rules employed by Brock et al. [25] disappears when the sample selection bias, data mining, hindsight bias, and other usual biases are accounted for. Park and Irwin [28] also confirm that most studies that do encounter superior profitability of TTRs are subject to various problems in their testing procedures, including biases, which should be addressed in order to provide conclusive evidence. In addition, Harvey and Lu [29] draw attention that seemingly successful trading strategies can be encountered by chance, and the "no–biases" assumption of traditional tools of statistical analysis no longer hold.

The first strong solution for the data-snooping danger, still seen as the standard method for adjustment, was proposed by White [30], and was based on results from Sullivan, Timmermann, and White [24]. The procedure, entitled White's Reality Check (RC) for data snooping, tests the null hypothesis that the best model does not have predictive superiority over a benchmark versus the alternative that the best model is over performing.

Afterwards, there have been some attempts in the literature to improve this methodology. Mainly, Hansen [31] maintains that the RC procedure can be affected by testing a large plethora of irrelevant rules, an issue that can be corrected by the "Superior Predictive Ability" (SPA) test. Further Bajgrowicz and Scaillet [32] introduce the false discovery rate

(FDR) as a new approach to data snooping and show that the economic value of TTRs that has been previously reported in the literature is no longer significant.

In this paper, we proceed to check the robustness of our results first by applying the popular Brock et al. [25] bootstrapping methodology (on a higher number of randomly generated series than employed by the original study and most others thereafter) and further, we correct for the data-snooping bias by following the most commonly used technique, namely, the RC procedure. This approach has the advantage of allowing easy comparison of results with previous related studies, and thus contributes to a higher relevancy of results.

## 3. Data and Method

### 3.1. Data

In the empirical modeling, we use daily spot prices of the two main grades of crude oil (Brent and WTI), as well as daily prices of a representative energy-traded fund, the Energy Select Sector SPDR® Fund or XLE. As XLE has the smallest trading history, to ensure comparability across markets, we set the same data window for the three time series, and hence data will span 1 January 1999 through 29 March 2021, or a total of 5686 daily observations for each energy market.

Brent North Sea Crude (also known as Brent crude oil) and West Texas Intermediate (known as WTI crude oil) are the most widely traded oil grades. Brent Crude is produced in the North Sea between Shetland Islands and Norway, while West Texas Intermediate is produced in the United States fields. According to the US Energy Information Administration, "sweet crude" refers to crude oil that has sulfur content of less than 1%, a category that Brent and WTI both fall under. Furthermore, both are less thick (or "lighter") than other types of crude oils, making them quicker to process and thus more appealing to manufacturers of petroleum products. Brent crude is the reference price for crude oil in Africa, Europe, and the Middle East, and it is assumed that Brent determines the value of around two-thirds of global crude oil production. Alternatively, West Texas Intermediate stands as the major oil benchmark for North America. As far as trading crude oil is concerned, Brent crude oil is listed on the New York Mercantile Exchange (NYMEX), a division of the Chicago Mercantile Exchange (CME), whereas Brent is listed on the electronic Intercontinental Exchange (ICE). As a result of their respective host markets, delivery locations vary by country in the case of Brent crude, which is traded internationally, while the main delivery location for physical exchange and price settlement for WTI is Cushing, Oklahoma. The price differential between Brent and WTI (which is a consequence, among others, of different transportation costs, of the supply and demand balance in different parts of the world, of geopolitical events, etc.) is called a spread.

The Energy Select Sector SPDR® Fund (XLE) mirrors the S&P 500's market-cap-weighted index of US energy companies. The Select Sector SPDR Exchange Traded Funds divide the S&P500 into nine industry categories, with XLE representing the energy sector. As a result, XLE is an investment vehicle that provides traders with a desired exposure to firms in the oil, gas, and consumable fuel industries, and related services.

Crude oil prices are obtained from the Federal Reserve Bank of St. Louis's (FRED) database, which collects data from the U.S. Energy Information Administration, while data for XLE are collected from Yahoo! Finance.

We argue that a separate investigation for a recent and relevant time period (the 2020–2021 COVID-19 pandemic) is not only more appropriate, but also more relevant to academics and investment practitioners. We base our hypothesis on previous empirical findings on the performance of TTRs on energy markets that show that the returns to technical strategies are not consistently strong for periods up to 2005 [18] or up to 2019 [23]. Thus, in order to take a closer look at the turbulent ongoing pandemic period, we will subset the so-called "COVID-19 window," which is spanning 1 January 2020 through 29 March 2021.

Figure 1 reflects the evolution of the BRENT and WTI crude oil price from January 2020 to March 2021, showing historical lows and significant volatility during the pandemic period. On 20 April 2020, the WTI crude oil price was disconnected from its typical relationship with the price of Brent crude oil, collapsing by more than \$50/barrel.

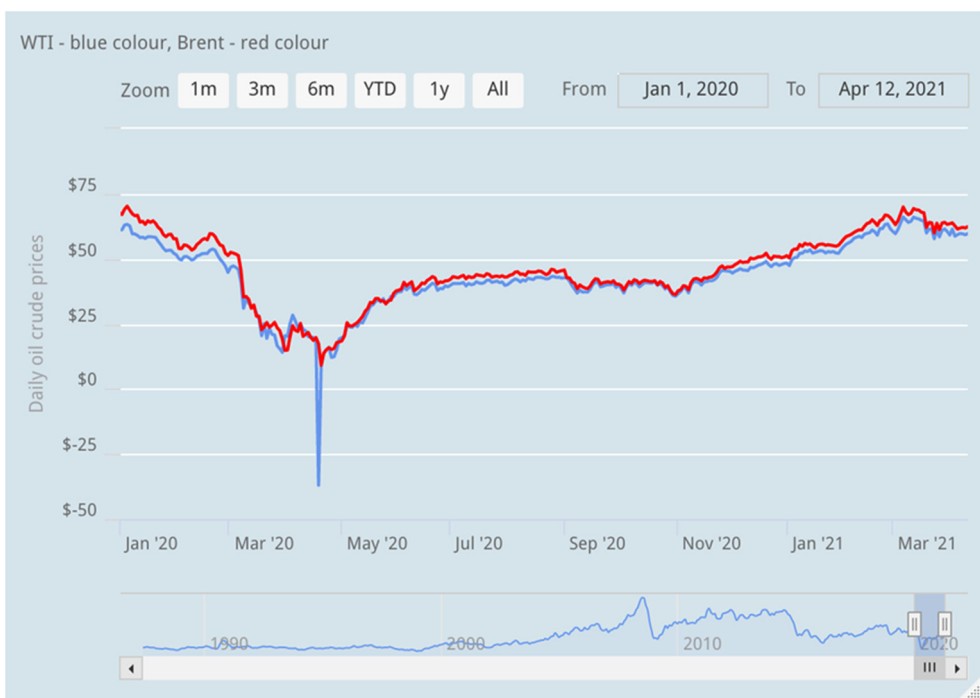

**Figure 1.** Spot Prices (Crude Oil in Dollars per Barrel) during the pandemic period (January 2020–March 2021). Source of data: U.S. Energy Information Administration, Crude Oil Prices: West Texas Intermediate (WTI)—Cushing, Oklahoma and Europe Brent, retrieved from FRED, Federal Reserve Bank of St. Louis; Author's representation.

Overall, the price of both WTI and Brent crude oil during the pandemic period registered a dramatic fall in the early stages of COVID-19 up until April 2020 and a subsequent recovery to pre-pandemic levels by March 2021, attesting the efficiency of interventions by oil-producing countries that have imposed supply caps, and also reflecting the optimism about post-pandemic economic recovery resulting from the progress of COVID-19 vaccine distribution worldwide.

The three daily energy series are turned into daily returns indexed from R to T, so that T = R + n − 1. We follow White [27] and compute daily returns as:

$$y_{i,t+1} = \frac{Index_{i,t+1}}{Index_{i,t}} - 1 \tag{1}$$

where $y_{i,t+1}$ is the return of the Index $i$ on trading day $t + 1$.

Figure 2 provides an overview of the three energy markets average return volatility over the 1999–2021 period, attesting the particularly dramatic month of April 2020, especially in the case of the WTI crude oil market.

The summary of descriptive statistics for one-day buy-and-hold returns for all three energy series employed in the empirical estimations, for the pre-pandemic period and also for the COVID-19 window, are presented in Table 1, panel A and panel B, respectively.

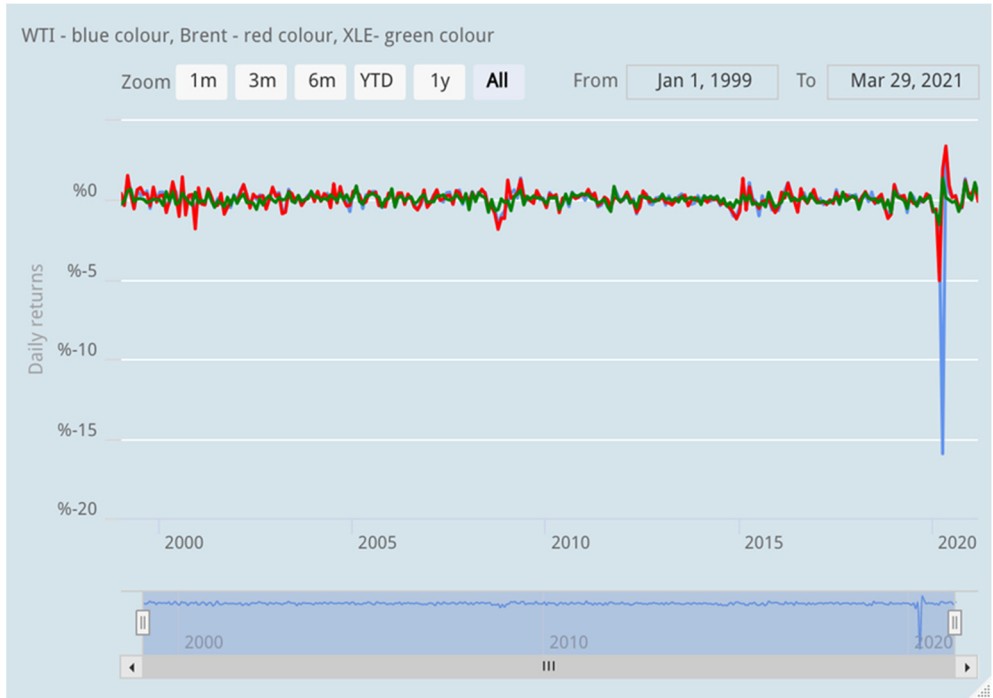

**Figure 2.** WTI, Brent, and XLE Average Monthly Returns (January 1999–March 2021). Source of data: Author's representation with crude oils daily price data sourced from the U.S. Energy Information Administration, retrieved from FRED, Federal Reserve Bank of St. Louis and XLE daily price data sourced from Yahoo! Finance.

**Table 1.** Descriptive statistics for one-day returns for WTI, Brent, and XLE.

| | Panel A: 1 January 1999–31 December 2020 | | | Panel B: 1 January 2020–29 March 2021 | | |
|---|---|---|---|---|---|---|
| | **WTI Crude** | **Brent Crude** | **XLE** | **WTI Crude** | **Brent Crude** | **XLE** |
| No of obs | 5367 | 5367 | 5367 | 319 | 319 | 319 |
| Min | −0.1571 | −0.1804 | −0.1444 | −3.0197 | −0.4747 | −0.2014 |
| Max | 0.1784 | 0.1988 | 0.1647 | 0.5309 | 0.5099 | 0.1604 |
| Range | 0.3355 | 0.3791 | 0.3092 | 3.5505 | 0.9845 | 0.3618 |
| Sum | 3.7387 | 3.3175 | 2.3881 | −2.8715 | 0.4065 | 0.0742 |
| Median | 0.0011 | 0.0006 | 0.0007 | 0.0021 | 0.0027 | −0.0009 |
| Mean | 0.0007 | 0.0006 | 0.0004 | −0.0090 | 0.0013 | 0.0002 |
| SE mean | 0.0003 | 0.0003 | 0.0002 | 0.0109 | 0.0036 | 0.0020 |
| CI. mean. 0.95 | 0.0006 | 0.0006 | 0.0004 | 0.0214 | 0.0070 | 0.0039 |
| Variance | 0.0006 | 0.0005 | 0.0003 | 0.0378 | 0.0041 | 0.0012 |
| SD | 0.0241 | 0.0224 | 0.0168 | 0.1944 | 0.0639 | 0.0353 |
| Coef. var | 34.5940 | 36.2955 | 37.6722 | −21.5959 | 50.1049 | 151.9894 |

The mean daily returns for the energy series largely confirm common perceptions of these markets. The XLE fund shows returns that compare rather well with the crude oil series during the whole 22-year period, and it also presents the lowest volatility of price returns both before and during the COVID-19 pandemic. On the other hand, during the pandemic period, WTI is the least rewarding in terms of return and also the riskier in terms of volatility among the three series. The Brent crude oil market has the highest mean returns for the pre-pandemic period (of about 0.07% per day) and also for the COVID-19 window (0.13% per day), while WTI is the only market that lost in terms of daily returns over the 2020–2021 period, whilst also being the most risky market. WTI statistics are surely strongly influenced by the historical plummet that the WTI price has suffered in April 2020. We notice from data presented in Panel B of Table 1 the dramatic aforementioned daily

drop of over 300% for WTI crude oil prices in April 2020, the largest one-day decrease in history.

This is further also more clearly reflected in Figure 3, showing returns volatility for the three energy markets during the pandemic period.

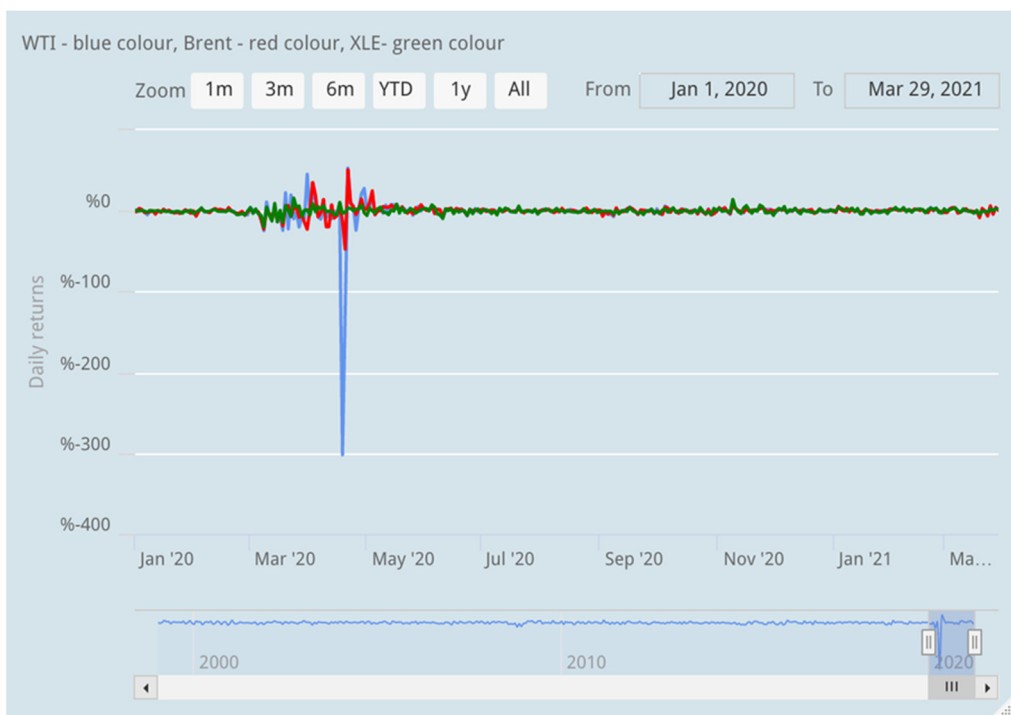

**Figure 3.** WTI, Brent Crude Oil and XLE Daily Returns during the COVID-19 pandemic (January 2020–March 2021). Source of data: Author's representation with crude oils daily price data sourced from the U.S. Energy Information Administration, retrieved from FRED, Federal Reserve Bank of St. Louis and XLE daily price data sourced from Yahoo! Finance.

*3.2. Method*

3.2.1. Signals and Excess Return of Simple Moving Average Strategies

The simple moving average (SMA) crossover is, by far, the most widely used among technical trading rules or TTRs [33]. The traditional simple moving average (SMA) rule issues buy (sell) signals that generate trades. When the short-period moving average rises above (or falls under) the long-period moving average by a pre-specified level or percentage (which is often set to zero in investment practice), buying (or selling) trades are initiated. As such, when the short-period moving average (*S*) exceeds the long-period moving average (*L*), a purchase signal is issued as follows:

$$\left[\sum_{\lambda=1}^{S} P_{t-(\lambda-1)}/S\right] > \left[\sum_{\lambda=1}^{L} P_{t-(\lambda-1)}/L\right] \Rightarrow \text{Buy at time } t \tag{2}$$

where $P_t$ is the price at time *t*, and the band equals zero.

Sell signals are generated when the short-period moving average (*S*) is below the long-period moving average (*L*):

$$\left[\sum_{\lambda=1}^{S} P_{t-(\lambda-1)}/S\right] < \left[\sum_{\lambda=1}^{L} P_{t-(\lambda-1)}/L\right] \Rightarrow \text{Sell at time } t \tag{3}$$

Further, excess returns over a given benchmark produced by a SMA TTR is estimated as:

$$\hat{f}_{t+1} = (1 + y_{t+1}S_1(X_{1,t}, \beta_1^*))/(1 + y_{t+1}S_0(X_{0,t}, \beta_0^*)) - 1 \tag{4}$$

where $S_1$ and $S_0$ are "signal functions" that take two permissible values, 1 for long trading positions and $-1$ for short trading positions. The signal value represents the total percentage of capital allocated at moment $t$ in a trading position, which further implies a 100% allocation of capital at any moment in this trading system.

The signal function converts indicators $X_{1,t+1}$ or $X_{0,t+1}$ and parameters $\beta_1^*$ or $\beta_0^*$ in Equation (4) into trading positions. The nominator in the above equation represents the SMA technical rule to be tested, while the denominator represents the benchmark. Here, the buy-and-hold (BH) strategy, a traditional benchmark strategy in portfolio management, is the benchmark of choice.

Average excess return for a particular TTR is then estimated as:

$$\bar{f} = n^{-1} \sum_{t=R}^{T} \hat{f}_{t+1} \tag{5}$$

The parameters in Equation (4) are the lengths of the two MA averages ($n_1$ for the short MA and $n_2$ for the long MA). See Anghel and Tudor [34] for more detailed information of signals and excess returns of SMAs.

Although some pairs of parameters are popular in the literature and in practice, we avoid pre-setting them and instead we run all rules using parameters ranging from 1–30 for *S* and 31–500 for *L* for the first subperiod, and parameters ranging from 1–15 for *S* and 16–120 for *L* for subsequent pandemic subperiod. We decide to restrict the parameter $n_2$ to a maximum value of 120 (representing approximately 6 months of trading) in the second subsample, which is consistent with practitioners' trading strategies based on TTRs (i.e., Menkhoff [17] showed that technical analysis is generally employed for trading decisions that do not exceed a horizon of 6 months). In the first subperiod, we permit a wider investigation and allow the second parameter to vary up to a maximum value of 500, which represents more than two years of trading.

Thus, for the larger pre-pandemic window:

$$n_1 \in \{1:30\}$$

and

$$n_2 \in \{31:500\}$$

and therefore we have a total number of SMA TTRs tested on 21 years of data corresponding to the pre-pandemic timeframe equal to: length ($n_1$) $\times$ length ($n_2$) = 30 $\times$ 470 = 14,100 for the parameter $\beta_l^*$ in Equation (4).

Subsequently, for the smaller pandemic interval:

$$n_1 \in \{1:15\}$$

and

$$n_2 \in \{16:120\}$$

corresponding to a total number of SMA TTRs tested during the 1 and $\frac{1}{4}$ years of pandemic timeframe equal to: length ($n_1$) x length ($n_2$) = 15 $\times$ 105 = 1575 for the parameter $\beta_i^*$ in Equation (4).

Hence, first, we test 14,100, and subsequently 1575 technical trading crossover rules based on Simple Moving Averages, computed as:

$$short\ SMA_t = 1/n_1 \sum_{t-n_1}^{t} X_t \tag{6}$$

$$long\ SMA_t = 1/n_2 \sum_{t-n_2}^{t} X_t \tag{7}$$

The function $S_1$ in Equation (4) will then dynamically convert into trading positions (long or short) according to the specified 14,100/1575 SMA TTRs.

R software was used to implement the method and perform estimations.

### 3.2.2. Robustness Checks

The first step in our estimations consists in computing average excess returns over the benchmark buy-and-hold trading strategy, as in Equation (5) produced by the 14,100/1575 SMA TTRs for each of the three energy markets (WTI, Brent and XLE) in the two sample periods (pre-pandemic and pandemic).

Secondly, the significance of excess returns produced by the 14,100/1575 SMA rules is tested.

In order to accurately accomplish this task, we should consider the high non-normality of the three energy markets. For non-normal distributions, the null hypothesis of normality could lead to serious inference errors when estimating classis statistical significance diagnostics. All three energy series are highly non-normal, presenting highly leptokurtic distributions (see Table 2). Although this is expected from daily returns, especially in the case of crude oil markets, results are nonetheless surprising and show a huge amount of excess kurtosis for all three markets, both pre and during the COVID-19 pandemic, but especially higher during the pandemic period. Leptokurtosis signifies that negative returns occur more often than positive returns, and estimations confirm this is indeed the case for the crude oil market (both WTI and Brent) and also for the energy fund XLE. Further, the Anderson–Darling (A–D) test is estimated to test the normality assumption for the three energy markets in the two sample periods. Results presented in Table 2 allow us to reject the null hypothesis of normality for all markets and all time periods.

**Table 2.** Distribution characteristics.

| | Panel A: Pre-COVID-19 Period (1 January 1999–31 December 2019) | | | COVID-19 Window (1 January 2020–29 March 2021) | | |
|---|---|---|---|---|---|---|
| | **WTI Crude** | **Brent Crude** | **XLE** | **WTI Crude** | **Brent Crude** | **XLE** |
| Skewness | 0.080747 | 0.101254 | −0.134112 | −12.38 | 0.50 | −0.37 |
| Kurtosis | 7.3788 | 7.5677 | 11.8093 | 186.9 | 28.7 | 9.3 |
| A–D Test | 5362 * | 5379 * | 5352 * | 214 * | 289 * | 273 * |

* significant at 1%.

Thus, to deal with non-normality in our data when testing for significance, we implement the popular bootstrapping methodology proposed by Brock et al. [25] in estimating *p*-values, under a random walk assumption for the distribution of returns [35] for all three energy series. As such, the null model is first fit to empirical data and its parameters are further estimated. The residuals are subsequently 1000 times randomly re-sampled (i.e., Brock et al. [25] generated 500 random series in their original study) and combined with the model parameters to generate random price series that will present the same characteristics as the original series. According to Brock et al. [25], the results do not differ significantly irrespective of which null model is employed (random walk, AR (1), GARCH-M, or EGARCH). Thus, for the null hypothesis, we continue with the random walk assumption in this study.

Hence, firstly we test whether the 14,100 SMA TTRs can generate excess returns for traders in the three energy markets during the pre-pandemic period. Further, after first estimating excess returns produced by the 14,100 trading rules for the three energy series in the 1999–2019 period, the bootstrapping methodology allows to compare the excess returns produced by a particular TTR applied to the real time series to excess returns that resulted from the empirical distribution, where the empirical distribution has been constructed by applying the same 14,100 trading rules to 1000 simulated time series with replacement under the null of a random walk. Thus, we sample with replacement from the original return series 1000 times for each of our original energy markets (WTI, Brent and XLE), obtaining 1000 simulated series or markets for each of the three real energy

markets, each simulated series having the same length as the original series (i.e., 5367 for the pre-pandemic period). We therefore produce three data frames each with dimensions (5367 × 1000) on which the significance of each of the 14,100 TTRs is tested. This implies that for each of the three energy markets, for the pre-pandemic timeframe, the 14,100 TTRs are first applied on the real time series of returns and subsequently on 1000 simulated return series for the respective energy market. Finally, the returns for each trading rule and the mean return across trading rules are estimated.

The procedure will then be replicated for the smaller COVID-19 window so that three simulated data frames with dimensions (319 × 1000) will be produced (where 319 is the number of observations of the original series and 1000 the number of simulated time series).

The average return $\overline{f}_b^*$ is thus obtained by applying the TTRs on the simulated series, where $b = 1, \dots , B$ is the number of the simulation from the total of $B$ simulations performed. Here, $B = 1000$.

Next, for the pre-pandemic period, results' significance is tested by comparing excess returns obtained on each of the three real energy markets to excess returns produced on the $3 \times 1000$ total simulated series of returns, each of length 5367. The main idea underlying this bootstrap methodology is that for a trading rule to be statistically significant at the $\alpha$ level, it must generate more revenue on fewer than 1% of the bootstrapped series than on the original series. The bootstrap $p$-value is then the percentage of times the buy-sell profit for the rule is greater on the 1000 random series than on the original series.

The same method is applied during the COVID-19 interval, where $3 \times 1000$ simulated series, each of length 319 have been produced.

Therefore, the estimated bootstrap $p$-value results from comparing the average real return $\overline{f}$ with the quantiles of average simulated returns $\overline{f}^* = \overline{f}_b^*, b = 1, \dots , B$. Hence:

$$B \ random \ bootstrap \ p\text{-}value = \frac{\sum_{b=1}^{B} 1_{\{\overline{f} < \overline{f}_b^*\}}}{B} \tag{8}$$

Finally, we account for the inherent data-snooping bias by following the standard Reality Check (RC) procedure for data snooping proposed by White [30].

White [30] develops the Reality Check Test applied to the best model (here, the best performing TTR) selected from a large sample of previously tested models. His algorithm consists in firstly computing the performance of the benchmark, which is expressed here as average excess return over the BH return. Thus, the first step consists in computing $\overline{f}_1$—the average excess performance of rule 1, followed by computing $\overline{f}_1^* = \overline{f}_{1,b}^*, b = 1, \dots ,$ $B$, which is a vector of length B (the number of simulations or bootstrapped samples, here, set again to 1000) containing the average excess performances on simulated (bootstrapped) time series, all for rule 1. Basically, up to this point, the procedure is identical to the earlier random bootstrap $p$-value estimation.

Next, White [30] sets $\overline{V}_1 = \overline{f}_1$ and $\overline{V}_{1,b}^* = \overline{f}_{1,b}^* - \overline{f}_1, b = 1, \dots , B$, so that the performance of rule 1 relative to the benchmark is tested by comparing $\overline{V}_1$ with the quintiles of $\overline{V}_{1,b}^*$. Similarly, for rule 2:

$$\overline{V}_2 = max\left\{\overline{f}_2, \overline{V}_1\right\} \tag{9}$$

and

$$\overline{V}_{2,b}^* = max\left\{(\overline{f}_{2,b}^* - \overline{f}_2), \overline{V}_{1,b}^*\right\} \tag{10}$$

where, as before, $b = 1, \dots , B$. In order to test whether the best of rule 1 and 2 is better than the benchmark, $\overline{V}_2$ is compared with the quintiles of $\overline{V}_{2,b}^*$.

Thus, there is a recursive process of testing whether the best model for the $k$th rule is superior to the benchmark, where $k = 3, \dots , l$ and $l$ is the number of rules to be tested (here, $l$ equals first 14,100 and subsequently, 1575 corresponding to the two sub-periods). The method thus implies comparing:

$$\overline{V}_k = max\left\{\overline{f}_k, \overline{V}_{k-1}\right\} \tag{11}$$

with the quintiles of:

$$\overline{V}_{k,b}^{*} = max\left\{\overline{f}_{k,b}^{*} - \overline{f}_{k}, \ \overline{V}_{k-1,b}^{*}\right\} \tag{12}$$

where *b* = 1, . . . , *B* for each of the *l* rules until a conclusion can be reached about the best performing trading rule.

Formally, Reality Check *p*-value could be expressed as:

$$RC\ p\text{-}value = \frac{\sum_{b=1}^{B} 1_{\{\overline{V}_{l} < \overline{V}_{l,b}^{*}\}}}{B} \tag{13}$$

## 4. Results and Discussion

In Table 3, we present the parameters and performance (excess returns over the benchmark BH returns) for the best performing TTRs encountered on the three energy markets across the two subperiods (results for the pre-pandemic period are presented in Panel A, while results for the pandemic period are reported in Panel B). Random bootstrapping *p*-values resulting from 1000 iterations, together with the number of signals generated by the optimal TTR are also presented. Note that transaction costs are not included in the first estimations.

**Table 3.** The best TTRs' parameters and performance with no transaction costs and BH returns as benchmark.

| | Panel A: Pre-COVID-19 Period: 1 January 1999–31 December 2019 Total No. of SMA TTRs Tested: 14,100 | | | COVID-19 Window: 1 January 2020–29 March 2021 Total No. of SMA TTRs Tested: 1575 | | |
|---|---|---|---|---|---|---|
| | WTI Crude | Brent Crude | XLE | WTI Crude | Brent Crude | XLE |
| Best Rule: SMA ($n_1$, $n_2$) | | SMA (5, 33) | SMA (27, 281) | SMA (3, 28) | SMA (12, 17) | SMA (5, 16) |
| Excess Return (%/day) | −0.1324 | 0.00782 ** | −0.0024 | 0.08244 | 0.5357 *** | 0.2762 |
| Excess Return (%, annualized [1]) | −28.38 | 1.99 ** | −0.52 | 20.00 | 284.32 *** | 83.45 |
| 1000 Random bootstrap *p*-value [2] | | 0.037 | 0.118 | 0.182(0.854 Rpc) | 0.064(0.38 Rpc) | 0.148 |
| No of Signals | | 201 | 22 | 8 | 16 | 18 |

** denotes significance at the 5% level, *** denotes significance at the 10% level. [1] To be more suggestive, daily returns have been annualized such that for every market: annual excess return = [(1 + daily excess return)^252−1]. The benchmark return is the buy-and-hold return. [2] This represents the random bootstrapping p-values resulting from 1000 iterations across the three energy markets and the two subperiods. Even without adjusting for data-snooping bias, this approach is nonetheless relevant not only for comparative purposes with previous studies, but also as it helps in identifying the total number of TTRs that are profitable prior to data-snooping bias adjustment. A tested TTR is statistically significant at the 5% level if excess returns on the 1000 random bootstrapped series exceed excess returns on the original series less than 5% of the time.

Results in Table 3 indicate that technical analysis appears to be significantly more profitable over the pandemic period than over the pre-pandemic period. Excess returns achieved by all 14,100 SMA crossover TTRs are negative in the pre-pandemic period for the WTI and XLE markets, indicating some small profits only for the Brent crude oil market (annualized excess return of the optimal TTR over the buy-and-hold benchmark return of about 2%, which is statistically strong, with a 1000 random bootstrap *p*-value of 0.037). Figure 4 reflects excess return for all 14,100 tested TTRs for the Brent market over the 21-years of pre-pandemic period. We chose to show only the Brent market as it is the only one for which some over-performing rules exist. It is obvious by looking at the chart below that only a small number of strategies are able to gain excess return over the benchmark BH strategy for the Brent market in the pre-pandemic period. Indeed, estimations confirm that only 7 rules out of the universe of 14,100 (or approximately 0.04%) are over-performing during 1999–2019.

We thus far conclude that none of the 14,100 moving average crossover TTRs can generate excess returns on the WTI and XLE markets, suggesting that the two energy markets are weak-form efficient over the 1999–2019 period with respect to these technical indicators. However, it seems that the same 14,100 rules were able to achieve statistically significant excess return, albeit rather small in magnitude, for the Brent market over the 1999–2019 pre-COVID-19 period, indicating this market might present weak-form inefficiency.

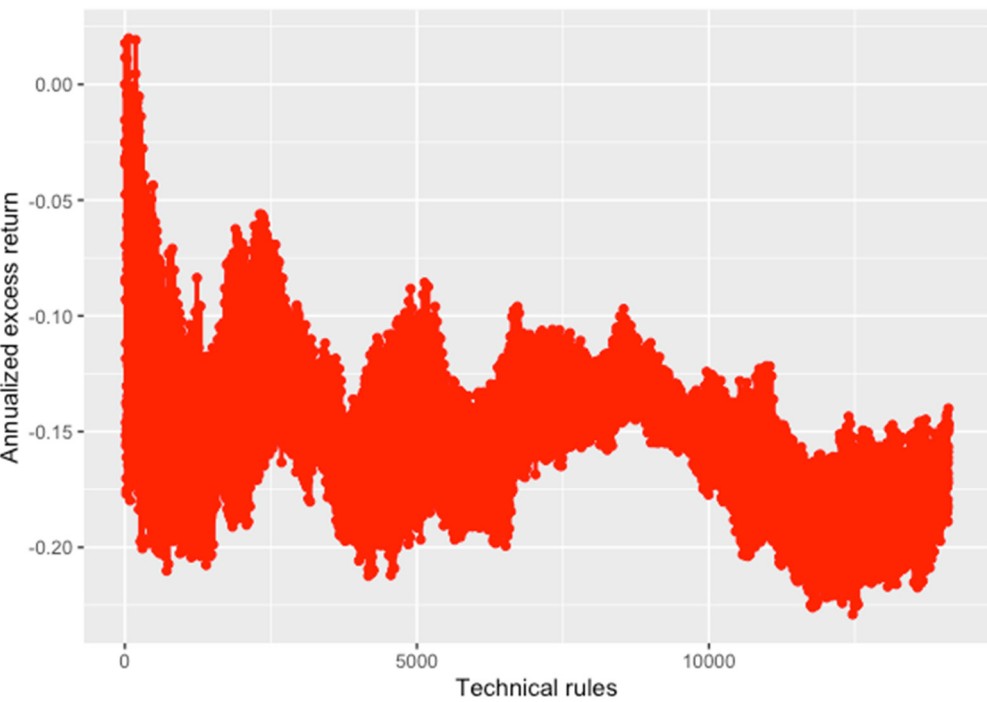

**Figure 4.** Annualized excess returns achieved by all 14,100 SMA crossover rules for the Brent market over the 1999–2019 period.

In turn, the pandemic period presents consistent excess returns achieved by the 1575 tested SMA trading strategies for all energy markets, and especially for the Brent market where an annualized excess return of over 284% has been achieved by the best performing TTR, which is SMA (12,17). However, for the WTI and XLE markets, this over-performance (annualized excess return of 20% for WTI and 83.45% for XLE) does not hold strong when its significance is tested via the standard bootstrapping methodology (1000 random $p$-values of 0.182 and 0.148, respectively). For the Brent market, TTRs are again able to achieve superior and statistically significant predictability (1000 random $p$-value equals 0.037).

Thus, we show in Figure 5 the excess return for all 1575 tested strategies for the Brent market over the 1 and $\frac{1}{4}$ year of pandemic period. Again, only the Brent market has been chosen, as it is the only one where signs of inefficiency are present. Therefore, while the best rule's performance indicates that over-performing trading strategies in terms of excess returns over the BH strategy exist for all markets in the second subperiod, the number of out-performing strategies is nonetheless very high for the Brent market. More precisely, 1528 out of the total number of 1575 TTRs (more than 97%) managed to achieve positive excess returns (which is also confirmed by Figure 5, where it can be easily seen that most of the strategies gain abnormal returns during the COVID-19 pandemic, whilst we remember that only 7 TTRs were found to be over-performing over the pre-pandemic period).

Moreover, as mentioned earlier, these excess returns are statistically significant for the Brent market during the pre-pandemic and also during the COVID-19 period (with 1000 random bootstrap $p$-values of 0.037 and 0.064, respectively). Moreover, we notice that during the pandemic period, the most successful SMA TTRs are the ones with shorter time horizons in the long-run moving average, while the time horizon for short-run moving average varies across the three markets. For example, $n_2$ equals 16 (XLE), 17 (Brent), and 28 (WTI) when it is allowed to vary in the interval (16:120) during the pandemic period.

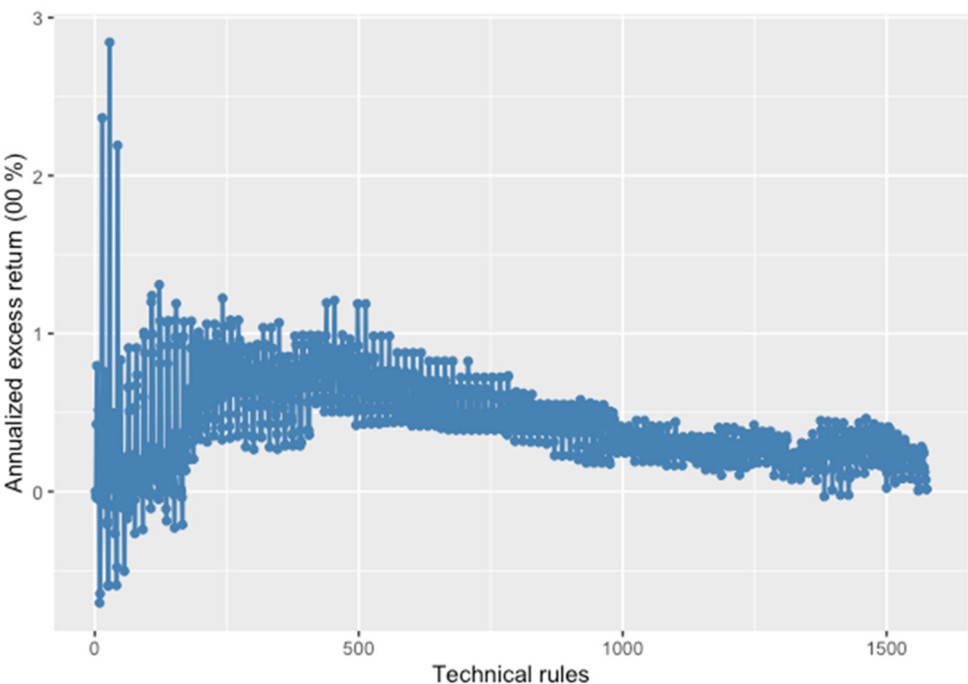

**Figure 5.** Annualized excess returns * for all 1575 SMA crossover rules for the Brent market over the January 2020–March 2021 COVID-19 pandemic period.

There is also some variation in the trading frequency of the best performing trading rule across the three energy markets over the two subperiods. For example, during the pandemic window in the WTI market, the most profitable TTR only signals a total of 8 trades over the 1 and $\frac{1}{4}$ year period, whilst for Brent and XLE 16 and, respectively, 18 trades are generated. Surprisingly, the best performing TTRs over the 1999–2019 period do not signal significantly more trades than over the much shorter pandemic period for WTI and XLE. On the contrary, for the Brent market, the optimal TTR is the short-term moving average rule SMA (5, 33), which generates a total of 220 trading signals over the pre-pandemic 21-year period, whilst the optimal TTR generates only 16 trading signals over the COVID-19 period, as seen above.

Despite the fact that the analysis is performed ex-post and also that transaction costs have not been included at this point, the above results still indicate some predictability of technical indicators in the case of Brent market, and especially during the pandemic period, that needs further investigation.

So, we test next for the economic significance of results and find that excess returns during the pandemic period remain abnormal for Brent when we include transaction costs in estimations. For the pre-pandemic window, excess returns disappear with the inclusion of trading costs.

Table 4 presents excess return net of transaction costs over the benchmark buy-and-hold strategy for the best performing TTR on the Brent market during the pandemic period, along with its corresponding 1000 bootstrapped *p*-value and the data snooping adjusted RC *p*-value. Meanwhile, Figure 6 reflects annualized excess returns net of transaction costs for all 1575 technical rules applied to the Brent market over the same period. The graph confirms that an overwhelming 96.20% of TTRs are still over-performing (1515 out of 1575 tested TTRs) after trading costs of 5 basis points (bps) are considered. This implies that only 13 rules' performance has been affected by the inclusion of transaction costs. Moreover, the over-performance is high in terms of magnitude of excess returns, with more than 92% of rules (1452 TTRs) achieving annualized excess returns of over 10%, and more than 31% of TTRs (493 rules) achieving annualized excess returns higher than 50%, while the best performing rule gains 270% annualized excess return net of transaction costs over the BH strategy. On the other hand, the under-performance is far less severe: only

60 TTRs out of the total universe of 1575 tested over the COVID-19 period (or 3.8%) have no economic value relative to the benchmark BH trading strategy, the majority of which (34 TTRs or 56.67%) under-performing by less than 10% in annualized terms.

**Table 4.** Excess returns of the best performing trading rule on the Brent market during the COVID-19 pandemic (January 2020–March 2021) net of transaction costs *.

| Optimal Trading Rule | No Signals | Daily Excess Return ** | Annualized Excess Return | 1000 Random Bootstrap *p*-Value | Reality-Check (RC) *p*-Value |
|---|---|---|---|---|---|
| SMA (12,17) | 16 | 0.5213% | 270.69% | 0.064 | 0.406 |

* Includes trading costs of 5 bps. ** The buy-and-hold strategy is the benchmark.

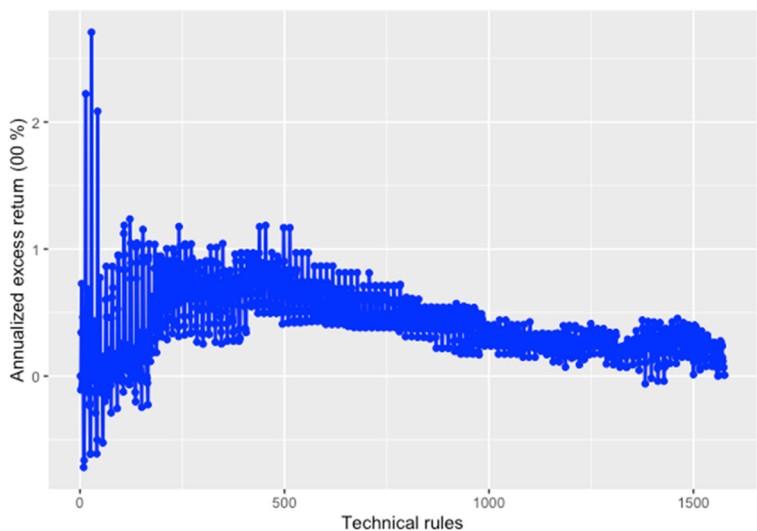

**Figure 6.** Excess returns* net of transaction costs of 5 bps for all 1575 SMA cross-over rules for the Brent market over the January 2020–March 2021 COVID-19 pandemic period.

When it comes to the statistical significance of the best TTR's performance, results hold strong when the bootstrapping methodology is applied (the 1000 random bootstrapping *p*-value of 0.064 is not affected by the inclusion of transaction costs in the estimation), but in turn the *p*-value resulting from the Reality Check test is no longer significant (RC *p*-value = 0.406). This indicates that the adjustment for data-snooping bias still has an important impact on the significance of results.

Overall, excess returns gained by the optimal TTR on the Brent market during COVID-19 do not hold strong after accounting for data-snooping bias by employing White's Reality Check test, but we feel the adjustment of the *p*-value via the RC procedure might be too severe and the procedure too conservative in this particular situation. We argue that the vast number of over-performing rules encountered on the Brent crude oil market over the COVID-19 pandemic period together with the magnitude of this over-performance, compared with the small number of underperforming rules (60 TTR out of 1575) and the "mild" relative underperformance (only 7 rules, or 0.04% of all TTRs achieve relative losses higher than 50%, while the majority encounter losses of less than 10% relative to the benchmark) already mitigates the data-snooping bias.

Consequently, in light of the aforementioned arguments, one cannot completely exclude the possibility that this adjustment via the RC procedure might be too severe and thus we should not be too quick to eliminate the possibility that over-performing TTRs might exist on the Brent market during the COVID-19 pandemic.

## 5. Conclusions

As a means of extending previous literature, this paper has analyzed the profitability of a significant number of SMA TTRs on a wider range of energy markets and a period that includes the ongoing COVID-19 pandemic.

Using daily data for Brent and WTI spot crude oil prices and for the energy fund XLE and splitting the data into a pre-pandemic window (1999–2019) and a "pandemic period" (January–March 2021), we employ 14,100 SMA crossover TTRs for the longer pre-pandemic period and 1575 SMA crossover TTRs for the shorter pandemic interval.

To the best of our knowledge, this is the first research attempt to investigate the effectiveness of technical indicators on both main crude oil markets, as well as on a relevant energy exchange traded fund, in comparative perspective between the pre-COVID-19 pandemic and the pandemic period.

Overall, we find that technical trading rules can achieve high abnormal returns for all three energy markets over the COVID-19 pandemic period (annualized excess returns over the BH strategy of approximately 20% for WTI, 284% for Brent, and 83% for XLE, without including transaction costs), and only for the Brent market, some small abnormal annualized excess returns (of about 2%) over the 21-years of pre-pandemic period.

However, these excess returns encountered over the pandemic period are not strong on the WTI and XLE markets when their significance is tested by the standard Brock et al. [25] bootstrapping methodology with 1000 iterations, while for Brent market, excess returns gained by TTRs hold strong in all subperiods against the standard bootstrapping methodology. Over the pandemic period, excess returns achieved by TTRs on the Brent market are still high in magnitude and remain statistically significant after transaction costs are included in estimations. Over the pre-pandemic period, the small excess returns achieved by some technical rules on the Brent market are eroded by transaction costs and thus have no economic value. Similar to Taylor [36], our findings could thus reflect a relationship between technical rules' performance and market conditions.

Nonetheless, the abnormal return achieved by the best-performing TTR on the Brent market over the 1 and $\frac{1}{4}$ years of pandemic period no longer holds strong against White's [30] Reality Check test. Thus, we find that SMA TTRs are not consistently profitable in the three energy markets once the data-snooping bias is accounted for.

However, while our results allow us to confidently conclude on the weak form efficiency of the WTI crude oil and the XLE fund markets throughout the 1999–2021 period relative to the universe of TTRs that we apply, and also to sustain the conclusion that TTRs do not add value on the Brent market beyond what may be expected by chance over the pre-pandemic 1999–2019 period, we refrain to also attest the weak-form efficiency of the Brent market over the COVID-19 pandemic. We feel that the performance of TTRs on the Brent market during the pandemic period needs further investigation, as most technical trading strategies achieve high excess returns over the benchmark buy-and-hold strategy, these excess returns hold when their significance is checked by the standard bootstrapping method and are also unaffected by transaction costs. The excess return gained by the optimal TTR only disappears after adjustment for data snooping is accomplished via the employment of White's Reality Check procedure. In this particular situation, the RC test might be too conservative and thus prone to type II errors. By presenting the results of all 1575 tested TTRs on the Brent market over the pandemic interval (both with and without transaction costs), and by showing that an overwhelming number of these strategies have been able to achieve abnormal returns of high magnitude on the Brent market during the COVID-19 pandemic (96.20% of strategies are still substantially over-performing even after adjustment for trading costs is made and thus have economic value), we sustain that survivorship bias is already mitigated. The adjustment made on the bootstrap *p*-values via the RC procedure could thus be too severe.

Consequently, while similar to Psaradellis et al. [23], we did not encounter enough evidence to be able to reject the weak-form efficiency of the three energy markets (Brent crude, WTI and XLE) for the whole 1999–2021 period, it would be hazardous to completely

dismiss the above argument in the case of the Brent crude oil market over the COVID-19 pandemic period, when TTRs seem to have benefited from the extreme evolutions that characterized the market. As such, future studies on the Brent market efficiency during crisis are needed to sustain the right policy formulation process.

This paper makes several contributions to the existing literature. The first contribution is to revisit the predictive ability and performance of technical trading rules (TTRs) on some oscillated energy markets. This is the first paper to include both main grades of crude oil (WTI and Brent crude) and a relevant energy fund, XLE, and to assess the trading rules' performance over two different subperiods: pre-COVID-19 (January 1999–December 2019) and COVID-19 (January 2020–March 2021). The second contribution thus consists in presenting proof of TTRs' performance during the historically turbulent COVID-19 pandemic period for crude oil markets. Previous studies mostly refer to the WTI crude oil and cover periods no more recent than year 2019. Other contributions consist in the large universe of tested TTRs (14,100 over the pre-pandemic period, and 1575 over the pandemic period, respectively) and also in estimating the relevancy of results by evaluating the performance of the universe of TTRs while considering both naïve [25] (Brock et al. random bootstrapping method—with 1000 iterations) and more severe methods of accounting for data snooping effects (White's Reality Check procedure—also based on 1000 iterations). In addition, this strategy allows easy comparison with previous findings that have employed one of the two (or both) techniques. A fifth contribution consists in also estimating the economic value of results by allowing for transaction costs, while a final contribution stems for the identification of distinct financialization process between the two main crude oil markets, WTI and Brent. As such, research findings further imply that there is evidence to the existence of a more intense financialization process within the WTI crude oil market, whereas the market for Brent seems to be more impacted by shifts in global supply and demand. This has important implications for both the right choice of oil price forecasting methods by policy issuers and also for identifying the accurate policy measures.

Policy makers must thus consider these market characteristics that the study encounters for effective oil price forecasts and for efficient policy issuance. Moreover, as the financialization of the WTI market and the approaching expiration date for WTI contracts for delivery in May 2020 [37], coupled with insufficient storage capacity have determined its historical and unforeseen plummet on April 2020 into uncharted negative territory [38] regulators of WTI commodity market (i.e., The U.S. Commodities Futures Trading Commission), should also consider tighter measures (i.e., mandatory reporting of high volume trades, short selling restrictions, etc.) to prevent the recurrence of such events.

**Author Contributions:** Conceptualization, C.T.; Data curation, C.T.; Investigation, C.T.; Methodology, A.A.; Software, A.A.; Validation, C.T.; Visualization, C.T.; Writing—original draft, C.T. Both authors have read and agreed to the published version of the manuscript.

**Funding:** This research received no external funding.

**Institutional Review Board Statement:** Not applicable.

**Informed Consent Statement:** Not applicable.

**Data Availability Statement:** Data for crude oil prices is publicly available from the Federal Reserve Bank of St. Louis's (FRED) database, which collects data from the U.S. Energy Information Administration, while data for XLE is publicly available from Yahoo! Finance.

**Conflicts of Interest:** The authors declare no conflict of interest.

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
