# Peer review of "The Financialization of Crude Oil Markets and Its Impact on Market Efficiency: Evidence from the Predictive Ability and Performance of Technical Trading Strategies"

_energies, doi:10.3390/en14154485_

Round 1

Reviewer 1 Report

This manuscript can be accepted after doing the comments.

Author Response

Thank you for these suggestions. All have been included/clarified in different sections of the paper, as follows:

- Figures numbers follow the order of figures in the paper (there was a material mistake in the pdf version of the original manuscript)

- English editing has been done.

Reviewer 2 Report

CONTRIBUTION: This work aims to provide a contribution towards a better understanding of the oil markets efficiency and financialization,  and possible impacts and implications of the COVID-19 

RECOMMENDATION: This study rises the interesting question. The methodological approach is sounding and findings interesting. I would suggest to better clarify aim and contributions of this study in the introduction, where only a few lines are deserved to define the scope of the study (lines 132-135). Also, how this study compares/differentiates to previous study should be better highlighted in the introduction and conclusions.  

The abstract should be shorter and better focused on novelty and results of this study. Equations, tables and figures should be better formatted to improve readability.

Also, consider to use shorter sentences (E.g. lines 282-288: This paragraph is a bit difficult to read).

Author Response

Thank you for these suggestions. They have been included/clarified in different sections of the paper, as follows:

- The presentation of the aim and contributions of the study in the introductory part has been extended, and also, how this study compares/differentiates from previous studies is now highlighted in the introduction and conclusions.  Thus the following paragraph has been underlined in the introduction: ”Nonetheless, and somewhat surprisingly given the practitioners’ interest in this commodity as reflected in its market liquidity, the academic literature on the profitability of technical trading rules applied to crude oil markets remains rather scarce. Our study contributes to extend this literature. This paper thus revisits the Fama efficiency of the crude oil markets through exploring the predictive ability and trading performance of a plethora of technical trading rules (TTRs) applied to relevant energy series (ie. WTI crude, Brent crude, and XTE). Moreover, our focus on energy/oil markets is even more motivated by the fact the COVID-19 pandemic has severely impacted the oil markets, due to travel restrictions, disrupted supply chains, and imposed government lockdowns. Previous studies have found that the efficiency of crude oil markets is lost during crisis periods, investigating the 2008 global financial crisis (Prokopczuk et al., 2019, Joo et al., 2020). As the impact of the ongoing pandemic crisis on oil market efficiency has not been yet assessed, this constitutes a secondary research goal of the current study and a further contribution to the extant literature.”

-and the following paragraphs have been underlined in the Conclusions:

To the best of our knowledge, this is the first research attempt to investigating the effectiveness of technical indicators on both main crude oil markets, as well as on a relevant energy exchange-traded fund, in a comparative perspective between the pre- COVID -19 pandemic and the pandemic period. “

“Similar to Taylor (2014), our findings could thus reflect a relationship between technical rules’ performance and market conditions.”

Consequently, while similar to Psaradellis et al. (2019), we did not encounter enough evidence to be able to reject the weak-form efficiency of the three energy markets (Brent crude, WTI, and XLE) for the whole 1999-2021 period, it would be hazardous to completely dismiss the above argument in the case of the Brent crude oil market over the COVID-19 pandemic period...”

- The abstract has been shortened accordingly, as follows: “Oil price forecasts are of crucial importance for many policy institutions, including the European Central Bank and the Federal Reserve Board, but projecting oil market evolutions remains a complicated task, further exacerbated by the financialization process that characterizes the crude oil markets.

The efficiency (in Fama’s sense) of crude oil markets is revisited in this research through the investigation of the predictive ability of technical trading rules (TTRs). The predictive ability and trading performance of a plethora of TTRs are explored on the crude oils markets, as well as on the energy sector ETF XLE while taking a special focus on the very turbulent Covid-19 pandemic period. We are interested in whether technical trading strategies, by signaling the right timing of market entry and exits, can predict oil market movements.

Research findings help to confidently conclude on the weak-form efficiency of the WTI crude oil and the XLE fund markets throughout the entire 1999-2021 period relative to the universe of TTRs. Moreover, results attest that TTRs do not add value on the Brent market beyond what may be expected by chance over the pre-pandemic 1999-2019 period, confirming the efficiency of the market before the year 2020. Nonetheless, research findings also suggest some temporal inefficiency of the Brent market during the 1 and ¼ years pandemic period, with important consequences for energy markets practitioners and issuers of policy.

Research findings further imply that there is evidence of more intense financialization of the WTI crude oil market, which requires tighter measures from regulators during distressed markets. The Brent oil market is affected mainly by variations in oil demand and supply at the world level and to a lesser degree by financialization and the activity of market practitioners. As such, we conclude that different policies are needed for the two oil markets and also policy issuers should employ distinct techniques for oil price forecasting.”

-The paragraph at lines 282-288 has been rephrased as: “We argue that a separate investigation for a recent and relevant time period (the 2020-2021 COVID -19 pandemic) is not only more appropriate but also more relevant to academics and investment practitioners. We base our hypothesis on previous empirical findings on the performance of TTRs on energy markets that show that the returns to technical strategies are not consistently strong for periods up to 2005 (Marshall et al., 2008) or up to 2019 (Psaradellis et al., 2019).”

Reviewer 3 Report

The manuscript clearly describes the research topic and presents a wide literature to support the comprehension to the reader. The methodology is well explained.

My suggestion is to shorten the abstract and perform an editing check for some minor typos and layout adjustments.

Here is my detailed review.

The manuscript investigates the performance and ability of technical trading strategies in predicting the behaviour of two Oil markets WTI and BRENT. Authors analyze data identifying two periods: a pre-pandemic period (1999 - 2019) and a pandemic period (2020 - 2021).
Strenghts:
- The manuscript has good scientific soundness and provides a wide literature to support the comprehension to the reader.
- The methodology of the analysis is well explained.
- The results and the differences between the two Oil markets are clearly presented.
Weaknesses: no relevant weaknesses identified
Recommendation: I suggest shortening the abstract, it is quite long for the Journal standards. There are some typos that need to be corrected and some layout adjustments required. Apart from these minor revisions, the manuscript is suitable for publication in my opinion.

Author Response

Response: Thank you for these suggestions. They have been included/clarified in different sections of the paper, as follows:

- The abstract has been shortened accordingly. It now reads:

“Oil price forecasts are of crucial importance for many policy institutions, including the European Central Bank and the Federal Reserve Board, but projecting oil market evolutions remains a complicated task, further exacerbated by the financialization process that characterizes the crude oil markets.

The efficiency (in Fama’s sense) of crude oil markets is revisited in this research through the investigation of the predictive ability of technical trading rules (TTRs). The predictive ability and trading performance of a plethora of TTRs are explored on the crude oils markets, as well as on the energy sector ETF XLE, while taking a special focus on the very turbulent Covid-19 pandemic period. We are interested in whether technical trading strategies, by signaling the right timing of market entry and exits, can predict oil market movements.

Research findings help to confidently conclude on the weak-form efficiency of the WTI crude oil and the XLE fund markets throughout the entire 1999-2021 period relative to the universe of TTRs. Moreover, results attest that TTRs do not add value on the Brent market beyond what may be expected by chance over the pre-pandemic 1999-2019 period, confirming the efficiency of the market before the year 2020. Nonetheless, research findings also suggest some temporal inefficiency of the Brent market during the 1 and ¼ years pandemic period, with important consequences for energy markets practitioners and issuers of policy.

Research findings further imply that there is evidence of more intense financialization of the WTI crude oil market, which requires tighter measures from regulators during distressed markets. The Brent oil market is affected mainly by variations in oil demand and supply at the world level and to a lesser degree by financialization and the activity of market practitioners. As such, we conclude that different policies are needed for the two oil markets and also policy issuers should employ distinct techniques for oil price forecasting.”

-English editing has been performed.